# Tozinameran (Pfizer, BioNTech) and Elasomeran (Moderna) Efficacy in COVID-19—A Systematic Review of Randomised Controlled Trial Studies

**DOI:** 10.3390/healthcare11111532

**Published:** 2023-05-24

**Authors:** Piotr Ratajczak, Zuzanna Banach, Dorota Kopciuch, Anna Paczkowska, Tomasz Zaprutko, Józef Krawczyk, Barbara Maciuszek-Bartkowska, Krzysztof Kus

**Affiliations:** Department of Pharmacoeconomics and Social Pharmacy, Poznan University of Medical Sciences, Rokietnicka 7, 60-806 Poznan, Poland

**Keywords:** COVID-19, tozinameran, elasomeran, BNT162b2, mRNA-1273, systematic review, meta-analysis

## Abstract

**Background:** The objective of this research was to test the efficacy and safety profile of tozinameran (30 μg, BNT162b2, Pfizer, BioNTech) and elasomeran (100 μg, mRNA-1273, Moderna) in COVID-19 prevention in ≥16-year-old patients vaccinated with two doses. **Methods:** A meta-analysis of the literature was conducted using the MEDLINE and EMBASE databases, following inclusion and exclusion criteria. Eight RCTs have been selected. The results were presented using the risk ratio (RR) with a 95% confidence interval (CI). A fixed-effect model or random-effect model was applied based on the heterogeneity of the results. **Results:** BNT162b2 and mRNA-1273 vaccines are efficient in preventing COVID-19 in comparison to a placebo (MH, RR 0.08 [0.07, 0.09] *p* < 0.00001 (95% CI)). It was found that administering the vaccines BNT162b2 and mRNA-1273 was associated with a higher proportion of adverse events in comparison to the placebo (IV, RR 2.14 [1.99, 2.29] *p* < 0.00001 (95% CI)). Administering the vaccines BNT162b2 and mRNA-1273 was associated with a higher proportion of serious adverse events in comparison to the placebo (MH, RR 0.98 [0.89, 1.08] *p* = 0.68 (95% CI)). **Conclusions:** Tozinameran and elasomeran are effective and safe in preventing the occurrence of COVID-19.

## 1. Introduction

In the face of the COVID-19 pandemic, vaccines containing tozinameran (Pfizer) and elasomeran (Moderna) were approved for use in December 2020 and January 2021, respectively [1,2]. Both preparations are mRNA based [3].

SARS-CoV-2 is one of the viruses under constant evolution. The essential variants demonstrating increased virus transmission and pathogenicity (as of the first half of 2022) [4] are:Beta (first diagnosed in the Republic of South Africa in September 2020),Gamma (first diagnosed in Brazil in December 2020),Delta (first diagnosed in India in December 2020),Omicron (first diagnosed in the Republic of South Africa and Botswana in November 2021).

In most cases, the characteristic symptoms of COVID-19 are similar to other respiratory diseases, making it difficult to diagnose and possibly enhancing the virus’s spreading [5]. 

Tozinameran (Comirnaty^®^, Pfizer, New York, NY, USA and BioNTech, Mainz, Germany), the Pfizer-BioNTech COVID-19 vaccine, is an mRNA-based vaccine developed by Pfizer and BioNTech to prevent COVID-19. The vaccine is highly effective in clinical trials, with an efficacy rate of 95% in preventing COVID-19 infections. The vaccine has been authorised for emergency use by regulatory agencies worldwide, including the FDA in the United States and the EMA in Europe. It has been administered to millions globally, and its safety and efficacy have been extensively studied and monitored. Its activity stimulates the immune system, conditioning the body’s response to potential contact with the SARS-CoV-2 virus. It enables the immune system to produce antibodies and blood cells to neutralise the virus and provide protection against COVID-19 [6]. Messenger RNA with modified nucleosides is enclosed in lipid nanoparticles; thus, RNA may enter the host cells and develop antigen S expression of SARS-CoV-2. RNA codes S-protein on the entire length with two point mutations in the centre spiral, and, as a result of the amino acid’s modification to proline, the S-protein becomes blocked in an antigen perforated prefusion conformation. As a result of the vaccine activity, patients develop an immune response—production of the neutralising antibodies and cell-mediated immunity—against the S-protein antigen [6].

Elasomeran, also known as the Moderna COVID-19 vaccine, is an mRNA-based vaccine developed by Moderna to prevent COVID-19. Like the Pfizer-BioNTech vaccine, it is highly effective in clinical trials in preventing COVID-19 infections. Like the Pfizer-BioNTech vaccine, elasomeran has been authorised for emergency use by regulatory agencies worldwide, including the FDA in the United States and the EMA in Europe [3]. It has also been extensively studied and monitored for its safety and efficacy. Elasomeran (Spikevax^®^, Moderna, Cambridge, MA, USA) has the form of dispersion for injections and is an mRNA vaccine (with modified nucleosides). The vaccine contains mRNA in lipid nanoparticles coding the S-protein of SARS-CoV-2 modified with two substitutes of prolines in the heptad repeat domain 1 (S-2P), which allows the S-protein to be maintained in the prefusion conformation. After injection, the lipid particles are absorbed and RNA is delivered into the cells to enable protein translation and biosynthesis. Following the expression, the immune system recognises the S-protein as a foreign antigen, which induces the immune response from T and B lymphocytes and promotes the production of neutralising antibodies.

Randomised controlled trials (RCTs) are widely regarded as the most reliable means of establishing cause-and-effect relationships in clinical research [7]. Evidence-based medicine (EBM) emphasises the importance of seeking the best possible evidence for each patient, and RCTs are generally considered the highest-quality study design in this hierarchy [8]. RCTs are typically used to objectively evaluate the impact of an intervention (such as a drug or procedure) on a particular outcome. While some researchers may argue that other sources of knowledge should also be considered, it should be noted that RCTs are designed to minimise the influence of extraneous variables and provide a high level of experimental control. As a result, they are considered a particularly robust means of establishing causality between interventions and outcomes.

The research objective of this study was to answer the question: Are tozinameran (Pfizer, BioNTech) and elasomeran (Moderna) efficient and safe in COVID-19 prevention in patients vaccinated with two doses? A quantitative assessment was performed regarding the vaccines’ efficacy about the number of patients who tested positive for COVID-19, and the safety profile based on the number of adverse events (AEs) (including serious adverse events SAEs) which were a consequence of the vaccines’ administration. To obtain the highest quality of results, a sensitivity analysis was also performed. This paper is based solely on randomised controlled trials in the field of COVID-19 vaccines.

## 2. Materials and Methods

### 2.1. Inclusion/Exclusion Criteria

The trials which met the following criteria were included in the review (inclusion criteria):Randomised controlled trial (highest quality of the data).Double-blind.The population covered by the trial included healthy persons aged over 16.Patients were vaccinated with two doses of tozinameran (30 μg) or elasomeran (100 μg) vaccines.The effects were compared to a placebo.The measure of the vaccine’s efficacy was the proportion of persons with or without a confirmed diagnosis of COVID-19 (based on RT-PCR test) 7 days after the 2nd dose of the BNT162b2 vaccine and 14 days after the 2nd dose of the mRNA-1273 vaccine.

Papers which met the following criteria were excluded from further analysis:
The trial did not include randomisation and double-blinding.Patients did not receive two doses of either vaccine.The comparator was not a placebo.The vaccine doses were other than 30 μg of tozinameran and 100 μg of elasomeran.

### 2.2. Databases and Publication Search Methods

The available literature was reviewed using the MEDLINE database via the PubMed and EMBASE databases. The selected keywords for the search are presented in the Appendix A. The MESH vocabulary thesaurus and EMTREE extended the searched topics. The terms of Boole’s logic AND and OR were applied during the search. The search was performed in February 2022 and included papers published since 2019. 

### 2.3. Bibliographic Analysis

A bibliographic data analysis was conducted on the number of published studies in the MEDLINE databse via the PubMed database related to the analysed vaccines (tozinameran and elasomeran). Studies were searched using keyword combinations defined in Appendix A, associating tozinameran or elasomeran with adverse events or outcome types. The number of RCTs, observational studies, meta-analyses, systematic reviews, case reports, clinical trials, and editorials in this pool were also determined by utilising the functionality available in PubMed. The time range of the publications covered the years 2020 to 2022.

The citation data of the studies included in the analysis were presented based on the results from the Web of Science website (as of 3 March 2023).

### 2.4. Data Selection and Collection

The selection of papers with the use of the inclusion/exclusion criteria and the website www.rayyan.ai was made by two independent researchers. The reviewers assessed the included publications and the conflicts were resolved by consultation. The respective phases of the selected papers are presented graphically with a PRISMA diagram [9] in Figure 1. 

The extracted data from each publication include the study’s first author, the number of participants in the trial group and control group, the dose of the vaccine, the number of doses, the type of diagnostics applied, and the method of reporting the adverse events. The literature review was based on the evidence from randomised clinical trials evaluating the efficacy and safety in the light of the application of elasomeran and tozinameran in persons over the age of 16. No limitations were found regarding sex, race, ethnic origin, cultural groups, publication language, trial region, or virus variant. Literature references given in the papers were also analysed to find additional publications. Additional limitations are presented in the Appendix A. 

### 2.5. Data Analysis

#### 2.5.1. Assessment of the Risk of Bias and Missing Data

The risk of bias was assessed for each included study based on the available methodology [10,11] using the Review Manager 5.4. software. The resulting conflicts were explained by consultation. No missing data were confirmed.

#### 2.5.2. Assessment of the Quality of Clinical Trials—JADAD Scale

Each RCT was assessed using the JADAD scale [12] to check its methodological quality (Appendix A). Each trial was scored on a scale from 0 to 5. Double-blind randomisation was given a score of 1, while single-blind was given a score of 0.5.

#### 2.5.3. Certainty of Evidence

Each reported outcome was assessed following the grading of recommendations, assessment, development and evaluation (GRADE) working group, using the GRADEpro tool (www.gradepro.org).

#### 2.5.4. Outcome Measurement

Outcome measurements conducted during the review process were:Vaccine efficacy was confirmed by the diagnosis of COVID-19 based on an RT-PCR test 7 days after the 2nd dose of the BNT162b2 vaccine and 14 days after the 2nd dose of the mRNA-1273 vaccine).The overall number of adverse events (AEs) such as erythema, tenderness, swelling, pain after injection, fever, headache, fatigue, myalgia, arthralgia, nausea, vomiting, and chills.The number of serious adverse events (SAEs) (life-threatening) such as hypersensitivity reactions, dermal filler reactions, Bell’s palsy, thromboembolism, and pericarditis.

#### 2.5.5. Quantitative Synthesis of Included Trials 

The data collected from the publications were entered into Review Manager 5.4. Afterwards, the data were verified by two authors. An MS Excel 2019 spreadsheet was also used for the initial stages of meta-analysis.

The analysis of dichotomous variables was presented with a risk ratio (RR) and the use of a 95% confidence interval (CI) [11,13]. The RR was calculated by dividing the risk of the outcome occurring in the intervention group by the risk of the same outcome occurring in the control group. A risk ratio of 1 means there is no difference in the risk between the two groups, while a risk ratio greater or lower than 1 indicates, respectively, a higher or lower risk in the intervention group compared to the control group. Due to the limited data, under the assessment of efficacy and safety profile of severe AEs the Mantel–Haenszel method and fixed-effect model were applied, as the analysed initial papers were found to demonstrate low heterogeneity—I^2^ < 50% (if the heterogeneity of the included results was lower then 50%, a fixed-effect model was applied). In contrast, the inverse variance method was used to assess the safety profile for the general number of AEs. This analysis was applied if the included studies demonstrated high heterogeneity (I^2^ > 50%), and, if so, the random-effect model was used. This is a model suitable for high heterogeneity because it specifies the wider confidence interval and allows the determination of the general effects measures compared to the fixed-effect model. 

The Mantel–Haenszel method used in the fixed-effect model involves calculating a weighted average of the risk ratios from each study, with the weights determined by the size of each study. The method considers any differences in the characteristics of the studies, such as study design or population, and provides a pooled estimate of the effect size that is adjusted for these differences. On the other hand, in the inverse variance method, each study’s effect size estimate is divided by its corresponding variance. The resulting value is known as the weighted effect size. The weights assigned to each study are the inverse of their variances. Studies with smaller variances, and thus more precise estimates, are given more weight in calculating the summary effect estimate.

A table summarising the initial trials included in the analysis and the meta-analysis results is presented in the Appendix A.

#### 2.5.6. Assessment of the Heterogeneity of the Studies

The collected studies contain variables concerning the methodologies used, the characteristics of patients, and the studied interventions. Such differences lead to the occurrence of heterogeneity. Therefore, after extracting the basic values subject to the trial, it was assessed whether the data was sufficiently homogeneous to combine it in the meta-analysis. 

The statistical heterogeneity was quantified with the use of the statistical test I^2^. The results were interpreted in compliance with the recommended methodology [11]. Forest plots generated using the Review Manager 5.4 software were also used in the heterogeneity assessment (visual interpretation). 

## 3. Results

### 3.1. Study Selection

In a review, 2202 publications were identified, of which 224 duplicates were rejected. Based on the titles and abstracts, 1911 studies were excluded due to the vaccines’ improper population and type or dose. Sixty-seven publications were selected for further analysis, and eight papers were included in the review after a full-text analysis. 

Studies not referring to the mRNA-1273 or BNT162b2 vaccines, or which did not contain data concerning the doses, 100 µg and 30 µg, respectively, were excluded from the analysis. Studies in which the results were compared to other vaccines without reference to a placebo were also excluded (Figure 1).

### 3.2. Characteristics of Studies Included in the Analysis

The review included eight studies. In two of them, patients aged over 16 were randomised [14,15]. Four studies covered patients aged over 18 [16,17,18,19]. In other studies [20,21], the trial group covered participants aged over 20. The pooled results are presented in Appendix A.

Patients with an immune disease reducing their immunity were excluded. All patients were administered the vaccine intramuscularly in two doses, BNT162b2 on days 0 and 21 and mRNA-1273 on days 0 and 28.

### 3.3. Assessment of the Quality of Studies Included in the Analysis

Each of the potential sources of bias was assessed as “high”, “low”, or “some concerns”. The assessment of the “risk of bias 2.0” for the studies included in the analysis is presented graphically (Figure 2). None of the studies was assessed as being burdened with a high risk of bias.

It was also assessed whether the trials were randomised, blinded, or the number of excluded participants was provided. The studies included in the meta-analysis were good quality (a score of 4.5 score out of a possible 5 on the JADAD scale) (Appendix A). 

The GRADEpro tool was used to assess the certainty of the evidence. The assessed outcomes for the BNT162b2 vaccine were clinical response, high quality; AEs, high quality; and SAEs, moderate quality (Appendix A). The corresponding outcomes for the mRNA-1273 vaccine were clinical response, high quality; AEs, low quality; and SAEs, moderate quality (Appendix A).

### 3.4. Results of the Meta-Analysis 

The efficacy (clinical response) assessment included two papers concerning the BNT162b2 [10,11] vaccine and two papers concerning the mRNA-1273 vaccine [12,14]. The AEs assessment included eight articles: four concerning BNT162b2 [14,15,19,20] and four concerning mRNA-1273 [16,17,18,21], which were additionally divided according to symptoms after the first and second doses of the vaccines. The eight previously mentioned papers were included in assessing the serious adverse events (SAEs): four concerning BNT162b2 and four involving mRNA-1273, but some of them [17,19,20,21] did not report any serious adverse events. 

### 3.5. Clinical Response

The results (Figure 3) show that both the BNT162b2 and mRNA-1273 vaccines were effective in preventing COVID-19 in comparison to the placebo (Mantel–Haenszel RR 0.08 [0.07, 0.09] *p* < 0.00001 (95% CI)). In total, four papers were included in the analysis. The number of patients was 67,617 for the tested intervention and 67,658 for the placebo. The total number of participants was 135,275. Statistical heterogeneity between the studies was low (I^2^ = 29%). The results for the subgroups: BNT162b2 (Mantel–Haenszel method RR 0.08 [0.07, 0.11] *p* < 0.00001 (95% CI), I^2^ = 61%); mRNA1273 (Mantel–Haenszel method RR 0.07 [0.05, 0.09] *p* < 0.00001 (95% CI), I^2^ = 0%).

### 3.6. Adverse Events

Based on the data obtained (Figure 4), it was found that administering the vaccines BNT162b2 and mRNA-1273 was associated with a higher proportion of AEs in comparison to the placebo (inverse variance RR 2.14 [1.99, 2.29] *p* < 0.00001 (95% CI)). The heterogeneity was 97% (I^2^ = 97%). The count was 104,087 for the intervention group and 103,555 for the placebo group. Two subgroups were introduced in the analysis: one concerned the BNT162b2 vaccine and the other the mRNA-1273 vaccine, the analysis of which was additionally divided into AEs occurring after the first and second doses, according to the reports on the trials. In both groups, a larger proportion of patients experiencing AEs existed in the group receiving the vaccine. Results for the subgroups: BNT162b2 (inverse variance RR 2.18 [2.12, 2,24] *p* < 0.00001 (95% CI), I^2^ = 0%); mRNA-1273 (inverse variance RR 2.13 [1.96, 2.32] *p* < 0.00001 (95% CI), I^2^ = 98%). The RR analysis for the events occurring after the first and second doses indicates a higher frequency of AEs after the second dose for both vaccines.

### 3.7. Serious Adverse Events

Based on the data obtained (Figure 5), it was found that administering the vaccines BNT162b2 and mRNA-1273 was associated with a higher proportion of SAEs in comparison to the placebo (Mantel–Haenszel RR 0.98 [0.89, 1.08] *p*= 0.68 (95% CI)). The total heterogeneity was 5% (I^2^ = 5%). The count was 74,409 for the intervention group and 74,189 for the placebo group. Results for the subgroups: BNT162b2 (Mantel–Haenszel RR 1.11 [0.93, 1.33] *p* = 0.23 (95% CI), I^2^ = 0%); mRNA-1273 (Mantel–Haenszel RR 0.93 [0.84, 1.04] *p* = 0.22 (95% CI) I^2^ = 0%).

### 3.8. Sensitivity Analysis

A sensitivity analysis was performed due to high heterogeneity (I^2^ = 97%) (Figure 4) in the total number of AEs. Since the medications’ manufacturers sponsored seven out of the eight papers in the analysis, no exclusion or sensitivity analysis concerning that fact was performed. 

### 3.9. Sensitivity Analysis Based on Two Doses of the Vaccines 

The initial division into subgroups concerning the BNT162b2 (results only after the second dose) and mRNA-1273 (results after the first and second doses) vaccines presented in Figure 4 was modified to subgroups concerning the analysis of the number of AEs observed separately after the first dose (only in the case of mRNA-1273) or after the second dose (BNT162b2 and mRNA-1273). 

Four studies were identified in which the assessment was based on only one dose [16,17,18,21]. In the other studies, only information on the number of adverse events after the second dose was available. After the analysis, the heterogeneity ratio for the observation of adverse events after the second dose of the vaccines was reduced from 97% to 40% (Figure 6), which means the reduction of heterogeneity to a moderate value (inverse variance RR 2.17 [2.17, 2.22] *p* < 0.00001 (95% CI)). Heterogeneity for the first dose was reduced, however, it still demonstrated a high value of 79% (Figure 6) (inverse variance RR 1.86 [1.78, 1.94] *p* < 0.00001 (95% CI)).

### 3.10. Sensitivity Analysis concerning Elasomeran

A sensitivity analysis of the number of AEs based only on the studies concerning elasomeran was also performed, as they demonstrated a higher level of heterogeneity (I^2^ = 98%). The initial division (Figure 4) was modified into four subgroups concerning the analysis of the number of AEs observed (Figure 7): the first group (I) was concerned with the result after the first dose [16,17,18,21]; the second group (II) concerned the result after the second dose [16,17,18,21]; the third group (III) included small trials [17,21]; and the fourth group (IV) included large trials [16,18]. After the analysis, the heterogeneity ratio was reduced for subgroups I, II, and III, however, it still indicated significant heterogeneity for subgroup IV (the heterogeneity increased to 99%) (Figure 7). The first dose (I) (inverse variance RR 1.86 [1.78, 1.94] *p* < 0.00001 (95% CI), I^2^ = 79%); the second dose (II) (inverse variance RR 2.18 [2.09, 2.26] *p* < 0.00001 (95% CI), I^2^ = 70%); small trials (III) (inverse variance RR 2.83 [2.24, 3.57] *p* < 0.00001 (95% CI), I^2^ = 60%); large trials (IV) (inverse variance RR 1.98 [1.81,2.18] *p* < 0.00001 (95% CI), I^2^ = 99%).

Higher heterogeneity occurred among large trials (99%) and in those papers which analysed the results after one dose (79%).

### 3.11. Publication Quantity and Citation Metrics

The analysis of the number of publications in the MEDLINE database via the PubMed database showed 4680 records related to tozinameran (88 studies were previously published as preprints) and 1643 records related to elasomeran (70 preprints). Among the records related to tozinameran, 693 papers were related to vaccine efficacy studies and 313 records covered adverse events. Regarding tozinameran, 45 RCTs, 225 observational studies, 32 meta-analyses, 75 systematic reviews, 635 case reports, 86 clinical trials, and 36 editorials were published. Most of the publications (4612 out of 4680) were in English. On the other hand, in the case of elasomeran, 269 records related to vaccine efficacy studies and 138 papers related to adverse events were found. A total of 30 RCTs, 68 observational studies, 24 meta-analyses, 55 systematic reviews, 250 case reports, 54 clinical trials, and 17 editorials were identified. Of the 1643 publications, 1620 were in English. The results are presented in Table 1.

On the other hand, the results regarding the level of citation of the studies included in the analysis indicate three studies with exceptionally high indicators, namely Polack et al. 2020 [14] (6825 citations, tozinameran), Baden et al. 2021 [16] (4566 citations, elasomeran), and Walsh et al. 2020 [19] (1312 citations, tozinameran). In addition, five out of the eight studies included in the meta-analysis were published in the *New England Journal of Medicine* (two were published in *Vaccine* and one in *Nature Communications*). The results are presented in Table 2.

## 4. Discussion

All of the studies included in the analysis were reliable (Figure 2 and Appendix A), however, the risk of bias could not be excluded [10,11]. A high risk of bias was not identified in any of the studies. However, six out of the eight studies had “some concerns” regarding their quality, as evaluated by the risk of bias 2.0 method. In most cases, the concerns were based on bias due to deviations from the intended intervention (authors mainly did not provide proper adherence to the intervention method [15,17,19,20,21]) as well as bias related to the randomisation process (no information about allocation sequence concealment) [14,15,19,20,21]. 

The meta-analysis has proven the efficacy of elasomeran and tozinameran in preventing COVID-19 in persons over 16 years of age compared to a placebo (Figure 3). It was confirmed that COVID-19 occurred in 0.22% of patients vaccinated with tozinameran, in comparison to 2.57% in case of the placebo. In studies concerning elasomeran, COVID-19 occurred in 0.23% of patients vaccinated with mRNA-1273 and in 3.29% in the placebo group (Figure 3). All of the included studies demonstrated significant results differences between the administration of the active substance and the placebo. In earlier meta-analyses, such a relation was also shown, but in most cases the authors were also concerned with the vector vaccines, which led to high heterogeneity. For example, in the paper by Fan et al. 2021 [22], the efficacy of mRNA, vector, and inactivated vaccines was combined in the analyses, and the result was favourable for the vaccines, however, heterogeneity amounted to 99%. With such high heterogeneity values, the results should be interpreted with caution. Fan et al. 2021 [22] included in their analysis ten studies, however, these also included non-randomised trials. This paper’s efficacy analysis involves four trials [14,15,16,18], all randomised and concerned with two vaccine doses. 

The results of the meta-analysis concerning the total number of AEs indicate that they were more frequent in vaccinated patients (Figure 4). When the analysed technologies examine vaccines, one must reckon with the occurrence of AEs resulting from the activation of the immune system as a result of which the specific immunity against a pathogen is developed. One must also take into account allergic reactions, for instance, in the result of contact with polyethylene glycol present in mRNA vaccines, and the fact that most trials were carried out in the autumn–winter season, therefore, the symptoms reported as post-vaccination might have been caused by contact with another virus, e.g., influenza or RSV. 

The analysis of serious AEs indicates that the analysed vaccines are relatively safe (Figure 5). In the group receiving a placebo, the occurrence of SAEs was less frequent in comparison to the group administered with BNT162b2, whereas in the placebo group, it was more frequent than in the group administered with mRNA-1723. This might be due to different qualifications of local or systemic events as SAEs, or non-reporting an event as serious if it was proven to be associated with the administered intervention. Additionally, in four out of the eight studies [17,19,20,21] no SAEs were reported, both for the vaccines and the placebo; therefore, they were not included in the analysis. 

General studies of the AEs indicated significant heterogeneity at the level of 98% (Figure 4), whereas the heterogeneity for the clinical response and SAEs analysis was 29% and 5%, respectively (Figure 3 and Figure 5). Multiple factors may cause this increased heterogeneity, e.g., the region in which the trial was made, the age of the participants, and the trial date, because, depending on the time when the study was performed, a different variant of SARS-CoV-2 virus was active. Moreover, the duration of each of the trials was different (Appendix A), and the follow-up of the participants lasted from 35 days [19] to a maximum of 225 days [18], which could be the reason for the more significant number of COVID-19 cases and AEs in the trials that lasted longer.

The duration of each trial also affects the determination of long-term immunity in patients vaccinated with two doses of the mRNA vaccine. Among the four studies included in the efficacy (clinical response) analysis, the longest was the trial by Sahly et al. 2021 [18], in which the patients’ follow-up period was 225 days. The most accurate method of determining the duration of immunity would be the analysis of the antibodies present in the blood of vaccinated patients. Four out of the eight studies included in the meta-analysis described the influence of vaccines on the level of antibodies by providing the values of GMT, however, the results were too heterogeneous to perform a proper analysis [17,19,20,21]. In trials concerning tozinameran, it was observed that the highest immunity occurred 7 days after taking the second dose [19] or was significantly increased on the 7th day after taking the second dose and maintained a similar value at least until day 35, marking the end of the results collection [20]. In the case of elasomeran, it was 14 days from the administration of the second dose [17,21]. This confirms that the method of case calculation assumed in this paper for the efficacy (clinical response) assessment (7 days after the second dose for the BNT162b2 vaccine and 14 days after the second dose for the mRNA-1273 vaccine) is correct.

Due to significant heterogeneity in the studies included in the analysis of general AEs (Figure 4), a double sensitivity analysis was performed. Statistical heterogeneity was quantified with the use of the statistical test I^2^. The first analysis was performed based on the number of vaccine doses (one or two doses regarding BNT162b2 and mRNA-1273). The second analysis concerned only elasomeran and was divided into four subgroups—results after the first dose, after the second dose, results from small trials and large studies. For the first analysis (Figure 6) there was a decrease of heterogeneity to a moderate value (40%) for the subgroup concerning AEs observed after the second dose of both vaccines. Moreover, in the case of the subgroup concerning AEs observed after only the first dose of elasomeran, a decrease in heterogeneity was also observed, however, the value was still too high (79%) (Figure 6). Therefore, the decision was made to perform a further sensitivity analysis, this time only for the vaccine based on elasomeran (Figure 7). For the subgroup of studies concerning the first dose, second dose, and small trials, the heterogeneity was reduced to 60–79%, whereas the results concerning large trials still demonstrated high heterogeneity (99%) [16,18] (Figure 7). So, the reduction of heterogeneity was only achieved in studies concerning events registered after the second dose in the analysis of both vaccines, probably because the analysed vaccines were described as two-dose interventions. The patient developed immunity against SARS-CoV-2 only after receiving an entire vaccination cycle. Reactions after the first dose were individual and dependent on the predisposition of each patient’s immune system. A high diversity of patients could cause high heterogeneity in large trials [12,14]. Moreover, in both studies, the intervention was administered to patients with a higher risk of having contact with the SARS-CoV-2 virus or patients with chronic diseases affecting the risk of complications due to COVID-19. Chen et al. 2021 [23], in their analysis, also achieved a high heterogeneity ratio for mRNA vaccine studies, which they justified with the ambiguities in the occurrence of allergic reactions associated with patients’ hypersensitivity to polyethene glycol (PEG) present in the vaccines. 

### Limitations

Despite a thorough search for publications in the MEDLINE and EMBASE databases, there is the risk that some available clinical trials or unpublished data were not identified. Additionally, all trials included in the analysis were short term (lasting 35 to 225 days), which limits the possibility of determining a long-term effect. 

Eight publications were included in the meta-analysis, but half concerned studies on relatively small groups. Although all publications were high quality and randomised, one must be cautious when applying the results to the general population, as small trials tend to overestimate the effect compared with large studies. The trials were short term, and each participant’s trial period differed. Thus, the median was applied to present the follow-up period. Patients with concurrent immune system diseases, chronic severe diseases, or unstable disease conditions were excluded from the trials, which may limit the possibility of referencing the results to a larger population. 

The analysis of adverse events was characterised by a high heterogeneity of the effect measure, which could be caused by differences in the vaccination schedule, taking two doses of the same vaccine, which differently engaged the immune system, and differences in recording the adverse events, which could deliver imprecise results.

While counting the events, the cumulative results were applied without specifying the populations of subgroups, such as elderly patients or patients particularly exposed to contact with the SARS-CoV-2 virus. The differences between sexes or participants from various ethnic groups were not analysed either. Additionally, there were differences in classifying events as serious, systemic, and local.

The trials were relatively short term, therefore, under the assessment of serious adverse events, no data was obtained that would apply to long-term trials, so the safety assessment may be insufficient to specify the potential events occurring within a longer period after receiving the vaccine.

Considering the severity and rapidity of the COVID-19 pandemic outbreak in 2020–2022, conducting clinical trials was much accelerated. The continuously emerging, new, more virulent virus variants may distort the assessment of the vaccines’ efficacy, depending on the period when they were performed.

Most of the trials included in the meta-analysis were financed by the manufacturers of the vaccines, therefore, one cannot exclude a conflict of interest that might have affected the reliability of the presented results. Seven out of the eight studies were sponsored by pharmaceutical companies manufacturing the vaccines (Pfizer/BioNTech, Moderna). One of the studies was supported by the pharmaceutical company Takeda.

## 5. Conclusions

It may be stated that both tozinameran and elasomeran are efficient in preventing COVID-19, however, the results must be referred to the general population with caution due to the short period of trials and the significant heterogeneity of publications included in the meta-analysis. Considering that the meta-analysis contained only eight randomised studies, further high-quality clinical trials must be carried out on the largest scale possible to verify and confirm the results.

## Figures and Tables

**Figure 1 healthcare-11-01532-f001:**
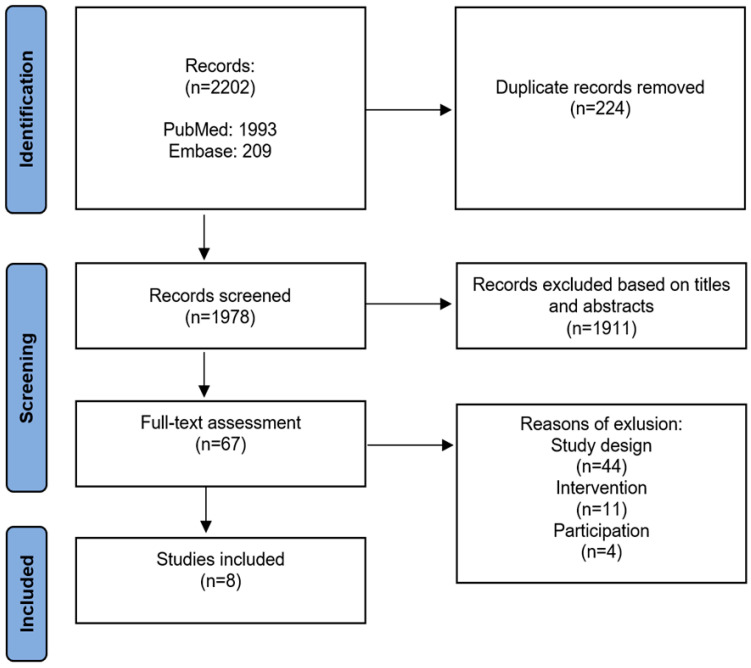
PRISMA diagram.

**Figure 2 healthcare-11-01532-f002:**
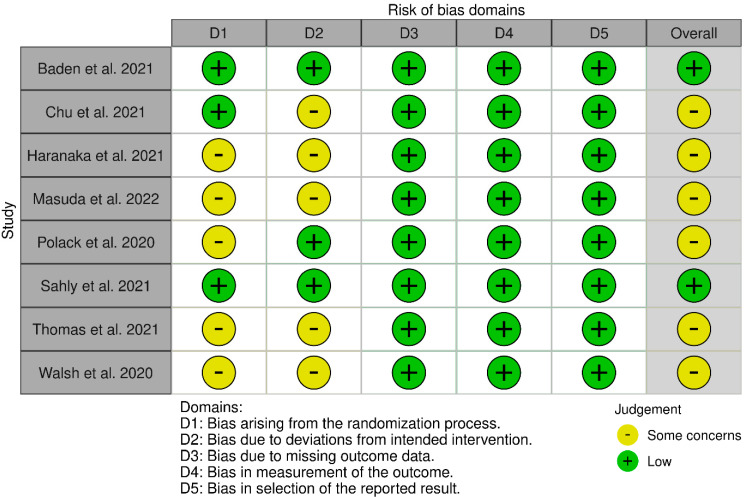
Risk of bias [14,15,16,17,18,19,20,21].

**Figure 3 healthcare-11-01532-f003:**
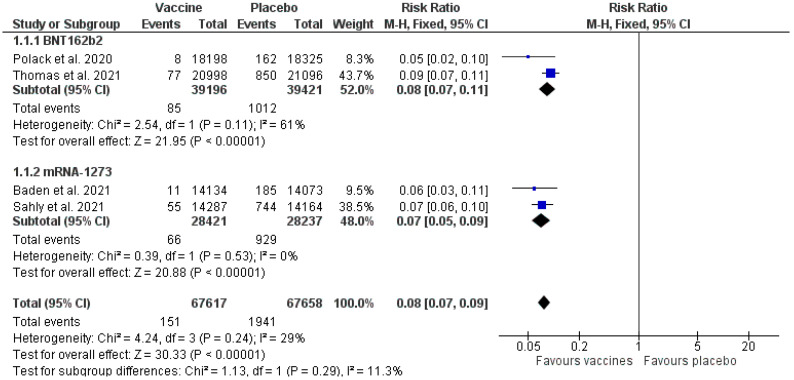
Clinical response—vaccines vs. placebo [14,15,16,18].

**Figure 4 healthcare-11-01532-f004:**
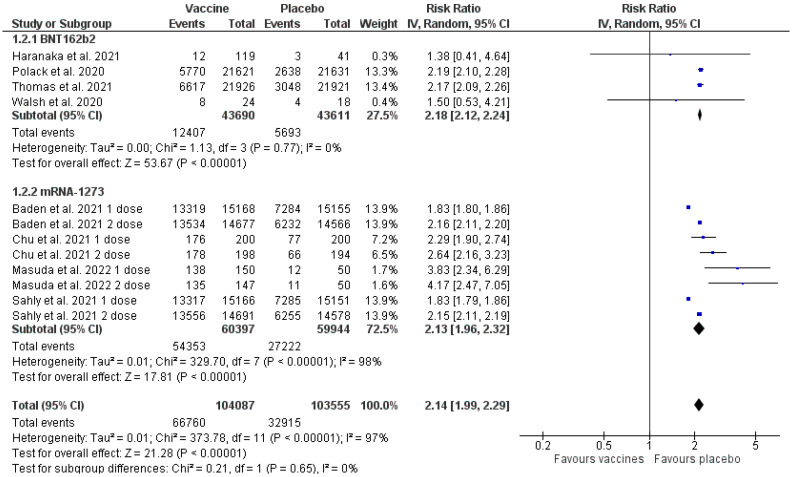
Adverse events—vaccines versus placebo [14,15,16,17,18,19,20,21].

**Figure 5 healthcare-11-01532-f005:**
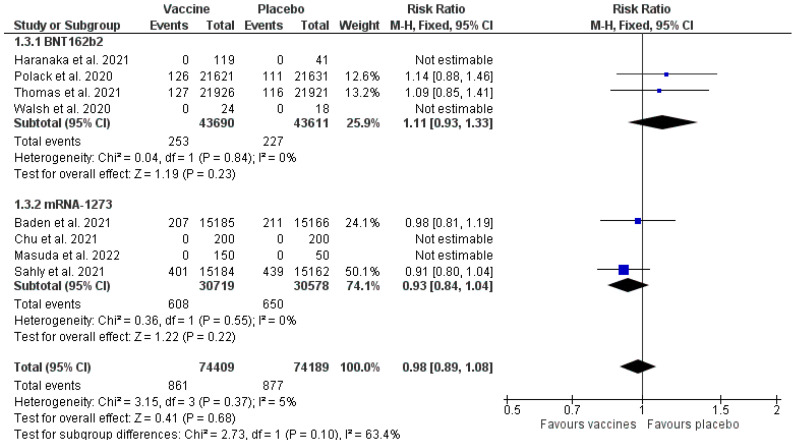
Serious adverse events—vaccines vs. placebo [14,15,16,17,18,19,20,21].

**Figure 6 healthcare-11-01532-f006:**
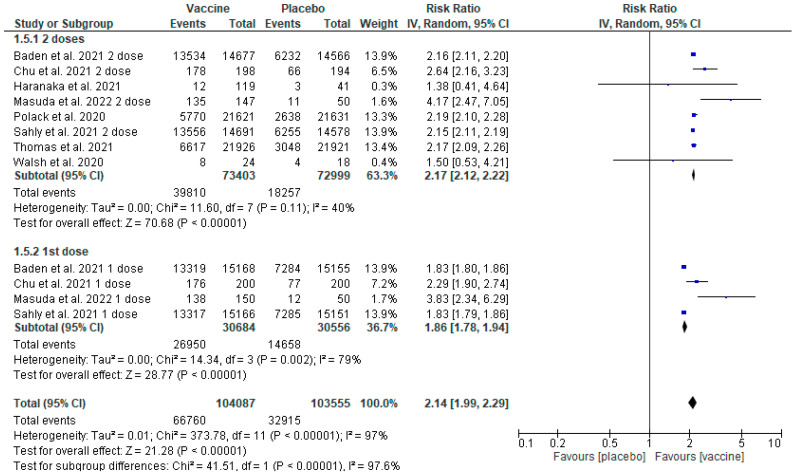
Sensitivity analysis based on doses (first subgroup: two doses (BNT162b2 or mRNA-1273); second subgroup: one dose (mRNA-1273) [14,15,16,17,18,19,20,21].

**Figure 7 healthcare-11-01532-f007:**
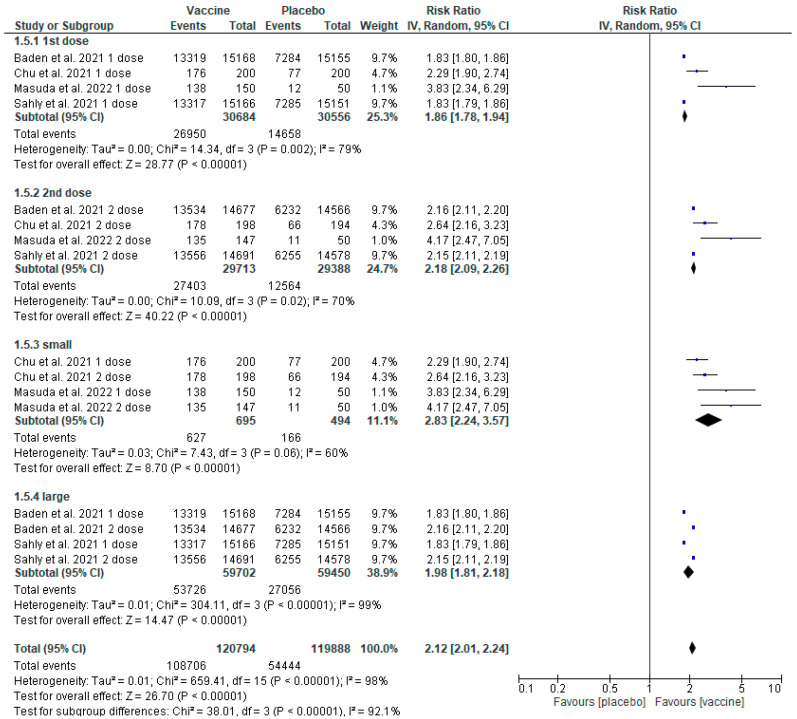
Elasomeran sensitivity analysis [16,17,18,21].

**Table 1 healthcare-11-01532-t001:** Analysis of the number of scientific papers in the MEDLINE database via the PubMed database regarding the analysed vaccines (2020–2022).

	MEDLINE via PubMed		Efficacy/Effectiveness Study	Adverse Events Study	Type of Study							Language
	Number of Papers	Number of Preprints			RCT	Observational	Meta-Analysis	Systematic Review	Case Reports	Clinical Trial	Editorial	English
Tozinameran	4680	88	693	313	45	225	32	75	635	86	36	4612
Elasomeran	1643	70	269	138	30	68	24	55	250	54	17	1620

**Table 2 healthcare-11-01532-t002:** Analysis of the popularity (citation metrics) of studies included in the meta-analysis.

	Type of Vaccine	Journal	Citation Number (Web of Science)
Baden et al. 2021 [16]	Elasomeran	*New England Journal of Medicine*	4566
Chu et al. 2021 [17]	Elasomeran	*Vaccine*	111
Haranaka et al. 2021 [20]	Tozinameran	*Nature Communications*	9
Masuda et al. 2022 [21]	Elasomeran	*Vaccine*	2
Polack et al. 2020 [14]	Tozinameran	*New England Journal of Medicine*	6825
Sahly et al. 2021 [18]	Elasomeran	*New England Journal of Medicine*	202
Thomas et al. 2021 [15]	Tozinameran	*New England Journal of Medicine*	462
Walsh et al. 2020 [19]	Tozinameran	*New England Journal of Medicine*	1312

## Data Availability

All data analysed during the study are available on reasonable request.

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
