# Peer review of "Tozinameran (Pfizer, BioNTech) and Elasomeran (Moderna) Efficacy in COVID-19—A Systematic Review of Randomised Controlled Trial Studies"

_healthcare, 2023, doi:10.3390/healthcare11111532_

Round 1

Reviewer 1 Report

Ratajczak et al. have done a systematic review of RCT studies on evaluating the efficacy and safety of tozinameran (Pfizer, BioNTech) and elasomeran (Moderna) COVID-19 vaccines in patients vaccinated with 2 doses.

This is an interesting review that focused on first selecting RCTs based on inclusion and exclusion criteria after conducting a meta-analysis of literature using MEDLINE and EMBASE databases. After the selection of 8 RCTs, quantitative assessments using extensive statistical analyses were done on the datasets to understand the clinical responses, adverse events and serious adverse events caused by the 2 vaccines vs placebo. Additionally, sensitivity analysis was also performed to obtain a high quality of results. This is a comprehensive and informative systematic review which has a few limitations owing to the availability of existing datasets. However, the following minor points need to be addressed to improve the quality of the paper:

Line 33: Abbreviations and their full forms should be written at the first appearance in the text.

Line 93: Should not be a bullet point and should be added separately

Figure 1: Please make sure all the words are legible. Example: ‘Screening’ in blue.

Tab. S4: Under ‘credibility assessment’ column, all commas (,) need to be replaced with decimal points (.) according to the described JADAD scale.

Tab. S5: In the second and last rows, all commas (,) need to be replaced with decimal points (.) according to the described JADAD scale.

Under section ‘3.5. Clinical response’, it is important to mention the criteria used to determine the efficacy of the 2 vaccines reviewed in the original studies.

The prior studies included in the review include information regarding the adverse events. It would be helpful to add a sentence that briefly mentions the adverse events that were evaluated in these studies under section ‘3.6. Adverse events’.

Under section ‘3.7. Serious adverse events’, please mention the serious adverse events evaluated.

Figure 6. It will be helpful to include which vaccine was studied here, 2 doses (BNT 162b2/mRNA-1273) and 1st dose (mRNA-1273).

Lines 313 and 314: For number of records, replace decimal point (.) with comma (,).

Table 1: Under number of papers, replace decimal point (.) with comma (,).

Table 2: Decimal point (.) needs to be replaced with comma (,) under citation number.

Line 403: Please change ‘80%’ to ‘79%’ to be consistent with the data in Figure 6.

Line 407: Please include ‘99%’ (which was the heterogeneity of large trials) in parentheses after this sentence to be consistent with the previous lines in the text.

Reviewer 2 Report

1.      Proofreading the manuscript is essential for avoiding writing errors.

  1. The authors should describe a little more about Tozinameran and elasomeran.
  2. Please describe in more detail about statistics determinations.
  3. Describe more details about publication quantity and citation metrics.
  4. The efficacy and safety profile of tozinameran and elasomeran in smokers and non-smokers were not determined.

Reviewer 3 Report

The manuscript by Piotr Ratajczak et al., they did a Meta analysis of literature to test the efficacy and safety profile of tozinameran  and elasomeran  in COVID-19 prevention in ≥16 years old patients vaccinated with two doses. The authors analyzed data from eight clinical trial research to investigate tozinameran  and elasomeran vaccine safety and efficacy. The data analysis showed both mRNA-1273 and BNT162b2 are very effective in preventing COVID-19 however, it was found that administering the vaccine was associated with adverse events and serious adverse events.  The authors used a proper methodology to sort the related articles to their interest and  the manuscript topics. The article is written in good scientific language and easy to understand the data. The data analysis has been done nicely using porper statistics softwares. The authors discussed the results in detail with supportive statements.  I have few comments to improve their articles before publishing   

1. I suggest not to use abbreviations in the title. Please use the full name of RCT for the first time and you can use the RCTs along your text after that.  2. Table

2. the citation number there is a typo error. I believe the number of citations is 4,566 and not 4.566 or 6,825 and not 6,825.  

3. I notice some typo errors through the text. Please go through the paper again to fix those typo errors
